# Comparison of Effluent Suspended Solid Concentrations from Two Types of Rectangular Secondary Clarifiers

Byonghi Lee

Department of Environmental Energy Engineering, Kyonggi University, Suwon 16227, Korea; bal@kgu.ac.kr; Tel.: +82-31-249-9737

**Abstract:** Secondary clarifiers play a significant role in the successful operation of activated sludge systems. Because of the restriction of available land, South Korean domestic wastewater treatment plants tend to employ rectangular clarifiers to settle mixed liquor suspended solids (MLSS) for activated sludge systems. A high MLSS concentration must be maintained in the bioreactor to ensure nitrification during winters, and achieve stringent effluent quality. The effluent suspended solid (SS) concentrations of two clarifier types currently being used in South Korea, primary rectangular clarifier-type and Gould Type I secondary clarifiers, were compared using computerized fluid dynamic simulations and hourly secondary effluent suspended solid concentrations. In addition, operational data such as hourly influent flow, daily MLSS concentrations, and sludge volume index were obtained and reviewed. This comparison reveals that the Gould Type I secondary clarifier is resilient to loading variation and produces effluent with a consistently lower suspended solid concentration than the primary rectangular clarifier-type under similar loading conditions and higher loading variations. The results suggest that the existing primary rectangular clarifier-type secondary clarifiers must be converted to Gould Type I.

**Keywords:** rectangular clarifier; Gould Type I; secondary clarifier

## 1. Introduction

Solid-liquid separation in secondary clarifiers plays an important role in the successful operation of suspended growth-activated sludge systems. The secondary clarifier is expected to provide high-quality supernatant along with thickened underflow, which is called return-activated sludge [1]. To achieve the stringent suspended solid (SS) concentration limits at final discharge, it is common to install filtration units after a secondary clarifier [2], and a secondary effluent with a low SS concentration has to be constantly provided for efficient operation of the filtration unit. When the SS concentration of the secondary effluent is high, the filtration unit must undergo frequent backwash operations. The backwash water thus produced must return to the headwork, adding extra flow and solid load to the mainstream process, in turn causing high strain during the wastewater treatment process. To achieve stable operation of the treatment process with a filtration unit, it is important that the secondary effluent has a consistently low SS concentration.

Several types of secondary clarifiers have been developed and used, including circular and rectangular clarifiers. There are two types of circular clarifiers based on feed location: center and peripheral feed clarifiers [3]. There are also two types of rectangular secondary clarifiers based on hopper locations [4]. When the activated sludge system was introduced in South Korea in 1970, the use of center-feed circular clarifiers was common. Since the 1980s, South Korea has experienced rapid urban growth, which demands efficient use of land for wastewater treatment plants. Because of the lower land requirement for rectangular clarifiers, they are being used as primary and secondary clarifiers since the 1990s. Although the solids settled in the primary clarifier are different from those settled in the secondary clarifier, the primary rectangular clarifier-type has been adopted as the

secondary rectangular clarifier, which is explained in detail in Section 2.1. No problems are known to occur in the secondary clarifier when a primary rectangular clarifier-type is used to settle mixed liquor suspended solids (MLSS) in a conventional activated sludge system, which maintains a bioreactor MLSS concentration of approximately 2000 mg/L.

In 1995, the South Korean government imposed a total nitrogen limit on the final effluent from wastewater treatment plants [5]. To remove nitrogen from wastewater, nitrification must occur in the aeration tank, and a high sludge retention time (SRT) must be maintained to ensure nitrification, especially in winters [6]. Because SRT is the ratio of total MLSS mass to that wasted within the system [7], a high system MLSS concentration is essential for a long SRT. For a conventional active sludge system that only removes organic matter, the system MLSS concentration is approximately 2000 mg/L. However, this concentration could reach approximately 3500 mg/L for nitrification, and this high MLSS concentration must be settled in the secondary clarifier.

Since 1995, almost all rectangular clarifiers in South Korea have experienced operational problems in winters. There are three issues with primary rectangular clarifier-type secondary clarifiers that settle MLSS from nitrification bioreactors. The most significant problem is rising sludge because of endogenous denitrification in the sludge blanket [8]. The risen sludge in the secondary clarifier becomes the effluent solid concentration in the secondary clarifier, which reduces the total MLSS mass and decreases the SRT. Eventually, nitrification will fail if the sludge rising continues. The second issue is varying secondary effluent SS concentrations. A sudden increase in the SS concentration is observed during a high inflow at the treatment facility. This increase results in a high solid load in the subsequent filtration unit and leads to the production of a final effluent with a high SS concentration. The third issue is the low MLSS concentration of the return activated sludge (RAS) due to clarifier shape. The settled MLSS in the secondary clarifier is collected in the hopper and sent back to the bioreactor as RAS. To achieve a high concentration of MLSS in bioreactors, the MLSS concentration of the RAS must be high. All three issues are correlated with SS concentration at secondary clarifier effluent, thus, the methods to produce constantly low SS concentrations must be investigated for the successful operation of the secondary clarifier.

The rectangular secondary clarifier with a hopper at the end is called Gould Type I [9], and one treatment plant in South Korea uses this type of clarifier. The operation data from this plant reveal that Gould Type I produces effluent with a constantly low SS concentration regardless of the variation in loading [10,11]. However, a comprehensive sampling is necessary to confirm these results, because weekly grabbed samples of secondary effluent are measured for SS concentrations in the previous report [10]. To compare the effluent SS concentrations from both the primary rectangular clarifier-type and Gould Type I clarifiers, effluents must be collected from both clarifiers at the same frequency. In addition, computational fluid dynamics (CFD) simulations must be performed with operational data, such as diurnal inflow, to confirm the experimental data.

## 2. Materials and Methods

Two wastewater treatment plants that use different types of rectangular secondary clarifiers were selected to compare the secondary effluent SS concentrations in winter. A high concentration of MLSS must be maintained in the bioreactor to sustain a long SRT in winter, which burdens secondary clarifier operation with high MLSS concentrations. To determine the difference in secondary effluent SS concentrations by clarifier type, the secondary effluent SS concentrations from each type were measured during the winter when the inflow to each clarifier had a high MLSS concentration.

The selected wastewater treatment plants include plant A and plant B, which use a primary rectangular clarifier-type and a Gould Type I rectangular clarifier as the secondary clarifier, respectively. To determine the effect of hourly inflow variation on the effluent SS concentrations from each clarifier type, secondary effluents were sampled hourly, and SS concentrations were measured on working days for two weeks. The corresponding hourly

inflow to each plant and daily RAS flow rates were provided by the personnel from each plant. In addition, each plant provided the daily MLSS concentrations and sludge volume index (SVI) of MLSS in the aeration tank during the sampling period. Although hourly secondary effluent samples from plant B were collected on working days for two weeks, the samples at plant A were collected sporadically. For plant A, nine 24-h hourly samples were collected from December 2019 to March 2020, and ten 24-h hourly samples were collected in January 2021, for plant B. An AQUAMATIC automatic sampler (Model P2-Multiform, Manchester, UK) was used for hourly sampling. The SS concentration of the secondary effluent was measured according to the Standard Method 2540 D [12].

CFD simulations were performed to determine the MLSS settling behaviors for each secondary clarifier type. A CFD package developed by McCorquodale et al. [13], which simulates fluid movement and MLSS settling, was used. The diurnal inflow, RAS flow, and MLSS concentration of the aeration tank, which supplied MLSS to the secondary clarifier, were used as input data for the CFD simulation. For both clarifier types, the same MLSS settling characteristics were applied to determine the difference in performance, including effluent SS concentrations.

### 2.1. Plant A—With Primary Rectangular Clarifier-Type Secondary Clarifier

Plant A is located south of Seoul, South Korea. It was commissioned in 2009, with a design treatment capacity of 47,000 $m^3$/day for the daily maximum and 37,600 $m^3$/day for the average daily flow. The mainstream process comprises the Bardenpho process [14], although the first aerobic reactor has media to which biomass is attached. This plant has four trains, each of which has two rectangular primary clarifier-type secondary clarifiers. Table 1 lists the design maximum, average daily inflow, and physical specifications for the mainstream process, including the secondary clarifier. Although Metcalf & Eddy/Aecom recommends a surface overflow rate (SOR) of 24–32 $m^3/m^2$·day for average daily flow [15], the design SOR is 18.4 $m^3/m^2$·day for average daily flow, which is well below the recommended range. Although solid loading rate (SLR) must be considered a major design element for secondary clarifiers, somehow, only SOR is regarded as the main design factor in South Korea. Figure 1 shows the longitudinal view and porous inlet wall of the clarifier used in the plant. MLSS from the second aerobic tank enters the secondary clarifier through the porous inlet wall. The wall has a width of 3869 mm and an effective height of 3595 mm. The diameter of each hole is 100 mm and each wall has a total of 143 holes. Based on this configuration, the opening ratio of the porous inlet wall is estimated to be 8%. As shown in Figure 1, the sludge scarper transfers the settled MLSS to the hopper, located just after the porous inlet wall. The sludge scarper moves against the direction of fluid flow on the clarifier bottom, which has a detrimental effect on the secondary effluent SS concentration.

**Table 1.** Design specifications of the primary rectangular clarifier-type secondary clarifier at Plant A.

| Item | | Values | Remarks |
|---|---|---|---|
| Design Flow | Maximum daily flow | 47,000 $m^3$/day | |
| | Average daily flow | 37,600 $m^3$/day | |
| Secondary clarifier | Number of units | 8 | |
| | Width | 8 m | |
| | Length | 32 m | |
| | Effective depth | 3.5 m | |
| | Total surface area | 2048 $m^2$ | |
| Design Surface Overflow Rate (SOR) | For maximum daily flow | 22.9 $m^3/m^2$·day | |
| | For average daily flow | 18.4 $m^3/m^2$·day | 24–32 $m^3/m^2$·day is recommended [15] |

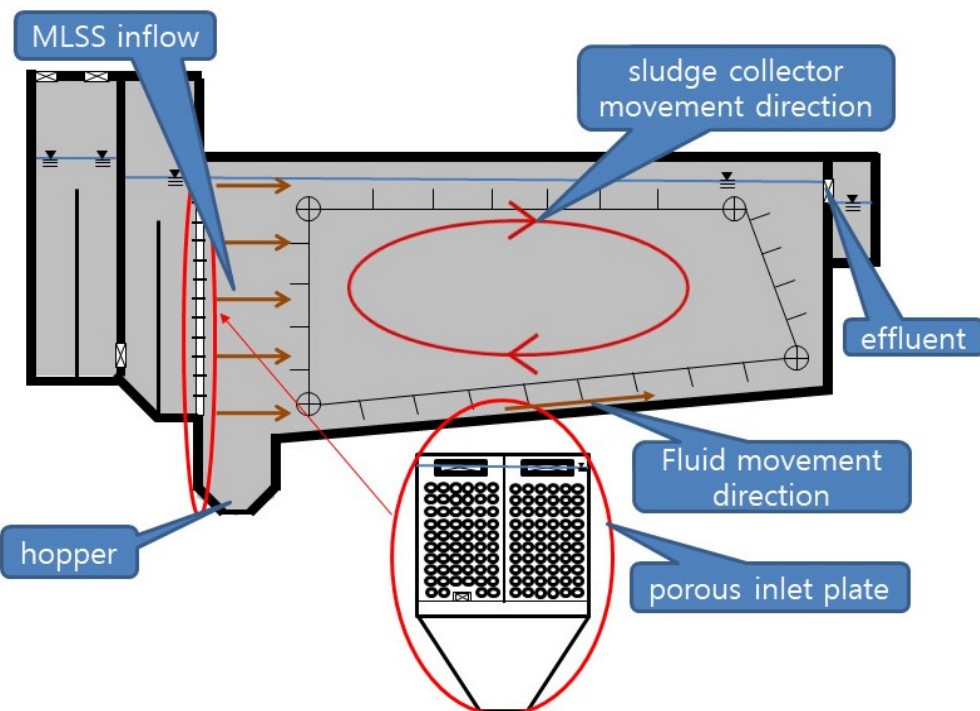

**Figure 1.** Longitudinal view of the secondary clarifier and porous inlet wall at plant A.

### 2.2. Plant B—With Gould Type I Secondary Clarifier

Plant B has been in service since 2010 and is located in northern Seoul. The design capacity of this plant is 14,000 $m^3$/day for the daily maximum flow and 11,200 $m^3$/day for the average daily flow. The main treatment process consists of two trains, each comprising pre-anoxic–anaerobic–anoxic–aerobic reactors, similar to the Johannesburg process [16]. Each train has a clarifier with 2 bays. A total of four bays of clarifiers are installed at the plant. The total surface area of the clarifier is 751.44 $m^2$ and the corresponding SOR for the daily average flow is 14.9 $m^3$/$m^2$·day, which is well below the recommended values by Metcalf and Eddy/Aecom [15]. Table 2 presents the physical specifications of the secondary clarifier at plant B, and Figure 2 shows the longitudinal view and energy dissipating inlet (EDI) of the Gould Type I clarifier. The physical aspects of the clarifier have been well documented [17,18]. In contrast to the primary rectangular clarifier-type, the hopper in Gould Type I clarifier is located at the end of the clarifier, and the sludge scraper transfers the settled MLSS in the same direction as the fluid flow at the bottom of the clarifier. The inflow to the clarifier must pass through the inlet diffuser, which is called the EDI, as shown in Figure 2. Three EDIs are attached to the front wall of each clarifier and the inlet pipe diameter of each EDI is 610 mm. The EDIs and hopper locations play important roles in producing effluents with constantly low SS concentrations.

When MLSS flow enters EDI, the plate at the end of EDI blocks the flow and MLSS spreads into four (4) outlet wings which are attached to the end plate of EDI. EDI converts MLSS flow direction from horizontal to vertical. Through this conversion, MLSS is spread along with the front wall of the clarifier. Normally there are three EDIs at each clarifier. The outlet wings of each EDI are aligned and MLSS from outlets at each EDI collide. Two EDIs located near the side walls of the clarifier send MLSS toward side walls through two (2) outlet wings of each EDI. EDI converts flow direction and spreads MLSS throughout the whole cross-section of the clarifier. Due to high velocity within EDI, no operational problem such as MLSS deposit is reported.

While EDI spreads MLSS along with the front wall, a porous inlet plate injects MLSS into the clarifier through holes. If porous inlet plate is used instead of EDI, high horizontal

velocity is expected at Gould Type I clarifier. For Gould Type I clarifier, MLSS must be entered through EDIs.

**Table 2.** Design specifications of the Gould Type I secondary clarifier at plant B.

| Item | | Values | Remarks |
|---|---|---|---|
| Design flow | Maximum day flow | 14,000 m$^3$/day | |
| | Average daily flow | 11,200 m$^3$/day | |
| Secondary clarifier | Number of units | 4 | |
| | Width | 6.2 m | |
| | Length | 30.3 m | |
| | Effective depth | 3.5 m | |
| | Total surface area | 751.44 m$^2$ | |
| Design SOR | For Maximum day flow | 18.6 m$^3$/m$^2$·day | |
| | For Daily Average flow | 14.9 m$^3$/m$^2$·day | 24–32 m$^3$/m$^2$·day is recommended [15] |

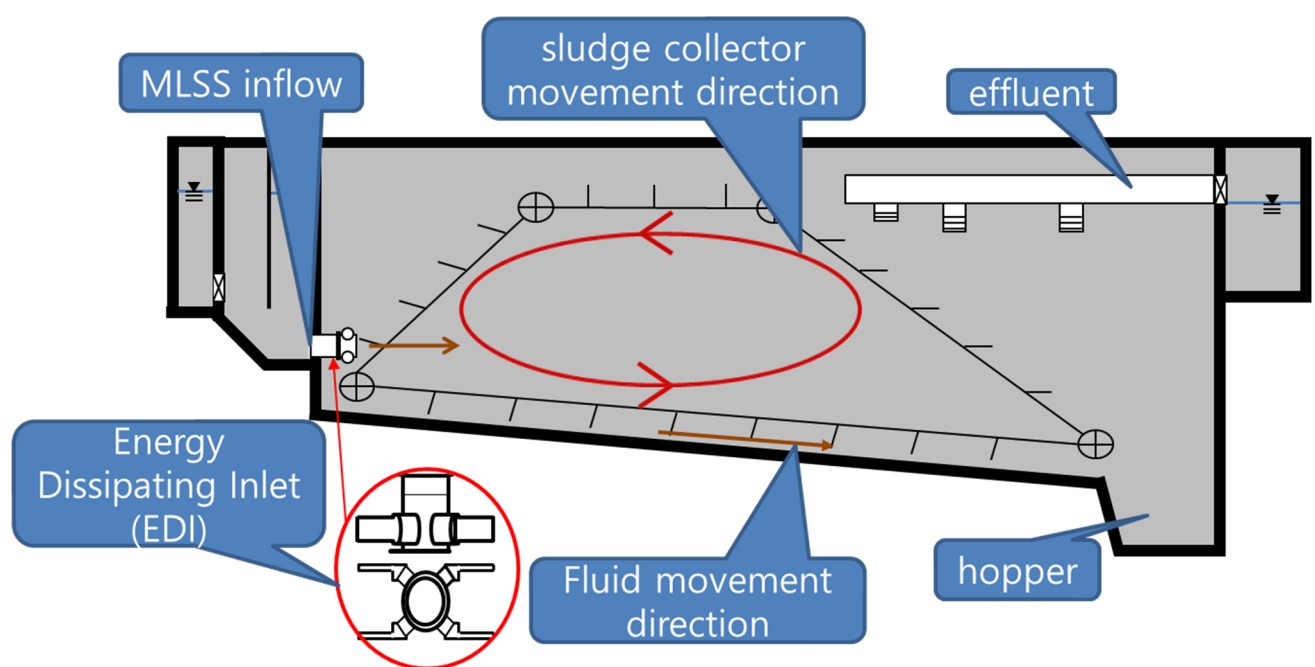

**Figure 2.** Longitudinal view of Gould Type I secondary clarifier and energy dissipating inlet (EDI) at plant B.

## 3. Results and Discussion

### 3.1. Experimental and Operational Data from Plant A

Supplementary Table S1 presents the diurnal secondary effluent SS concentrations and the corresponding hourly inflow at plant A, and Figure 3 shows the diurnal variation in SS concentrations. From the table and figure, it can be noticed that SS concentrations are highly variable and each sampling day has a sudden rise in SS concentration, except for 21 January 2020. To determine the cause of diurnal variation and peak SS concentration, the average hourly SS concentration was calculated, and the impact of loading on the SS concentration was investigated. Table 3 presents the daily MLSS concentration and RAS flow ratios of the influent during the sampling period. From these data and the hourly average inflow presented in Table 4, the hourly SORs and SLRs are calculated based on the

total surface area of the clarifier, which is 2048 m². Since MLSS concentration is changed daily, an average MLSS concentration from the beginning and ending days of sampling is used for the SLR calculation. Table 4 shows the estimated average hourly secondary effluent SS concentrations, influent flow rates, SORs, and SLRs for each sampling hour. The hourly influent flow is divided by the total surface area to calculate the SOR. Because the RAS pump is adjusted daily, the hourly RAS flow is assumed to be constant throughout the respective sampling day. For example, the hourly RAS flow is estimated at 849.75 m³/h on 9 January 2020, which is calculated as the RAS flow of 20,394 m³/day divided by 24 h. The total hourly inflow to the secondary clarifier is the sum of the influent flow and the estimated RAS flow for the respective sampling hours. The daily MLSS concentration in the bioreactor, which is measured by the operation staff on the sampling day, is to calculate the solid loading. Figure 4 presents the average hourly secondary effluent SS concentration at each sampling hour, along with the corresponding estimated hourly average SORs and SLRs during the sampling period. Figure 4 reveals that the hourly SORs change in the same pattern as hourly SLRs, and secondary effluent SS concentrations are highly related to SORs and SLRs.

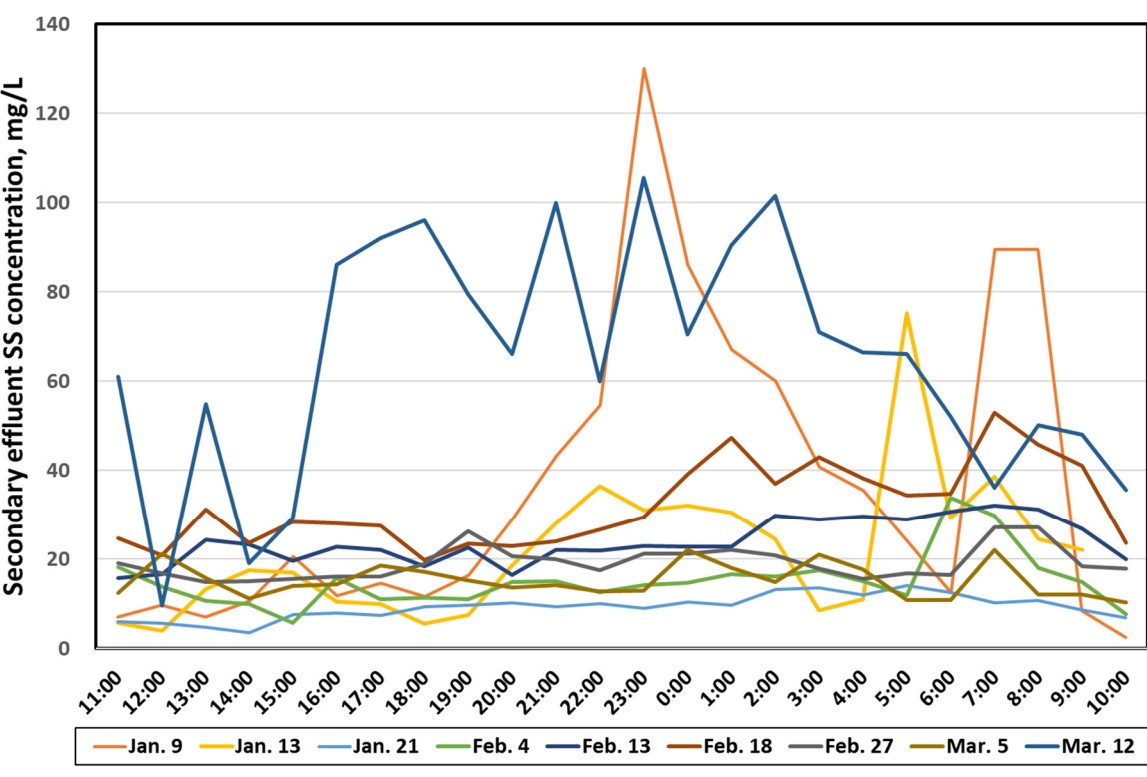

**Figure 3.** Diurnal variation of secondary effluent SS concentrations at the secondary clarifier at plant A during the sampling period.

The hourly changes in the secondary effluent SS concentration at plant A can be explained by the configuration of the secondary clarifier. Plant A has a primary rectangular clarifier-type secondary clarifier, in which the direction of fluid flow is opposite to that of the sludge scraper at the bottom of the clarifier. This opposite direction of fluid against that of the settled MLSS prevents the rapid removal of settled MLSS and creates a settled MLSS blanket at the bottom of the clarifier. The geometric characteristics of the primary rectangular clarifier-type secondary clarifier cannot avoid settled MLSS blankets, and endogenous denitrification within this blanket causes sludge to rise. The risen settled MLSS flocs are released as SS in the effluent. In addition, the increasing inflow rate affects the secondary effluent SS concentrations. Whenever there is a settled MLSS blanket in the

clarifier, the secondary effluent SS concentration increased rapidly when the increased inflow rate is sufficiently high to push the settled MLSS to the end of the clarifier.

**Table 3.** Daily inflows, RAS flows, RAS ratios, and MLSS concentrations on respective sampling days at plant A.

| Date | Inflow m³/Day | RAS Flow m³/Day | RAS Ratio % | MLSS Conc. mg/L | Average MLSS Conc. mg/L |
|---|---|---|---|---|---|
| 9 January 2000 | 38,970 | 20,394 | 52.3 | 3738 | 3750 |
| 10 January 2000 | 37,280 | 22,344 | 59.9 | 3763 | |
| 13 January 2000 | 36,670 | 21,512 | 58.7 | 3808 | 3790 |
| 14 January 2000 | 36,880 | 19,383 | 52.6 | 3773 | |
| 21 January 2000 | 36,350 | 18,833 | 51.8 | 3688 | 3746 |
| 22 January 2000 | 35,580 | 19,005 | 53.4 | 3805 | |
| 4 February 2000 | 35,750 | 19,640 | 54.9 | 3668 | 3713 |
| 5 February 2000 | 34,640 | 20,370 | 58.8 | 3758 | |
| 13 February 2000 | 33,840 | 18,340 | 54.2 | 3703 | 3680 |
| 14 February 2000 | 34,210 | 18,829 | 55.0 | 3658 | |
| 18 February 2000 | 34,650 | 19,409 | 56.0 | 3668 | 3684 |
| 19 February 2000 | 34,900 | 20,501 | 58.7 | 3700 | |
| 27 February 2000 | 34,770 | 20,137 | 57.9 | 3618 | 3563 |
| 28 February 2000 | 34,210 | 19,492 | 57.0 | 3508 | |
| 5 March 2000 | 34,100 | 19,059 | 55.9 | 3550 | 3545 |
| 6 March 2000 | 33,630 | 18,554 | 55.2 | 3540 | |
| 12 March 2000 | 34,140 | 18,887 | 55.3 | 3523 | 3524 |
| 13 March 2000 | 34,100 | 18,316 | 53.7 | 3525 | |
| Average | 35,259 | 19,611 | 55.6 | 3666 | 3666 |
| Maximum | 38,970 | 22,344 | 59.9 | 3808 | 3790 |
| Minimum | 33,630 | 18,316 | 51.8 | 3508 | 3524 |

Note: Conc. is concentration.

The hourly average secondary effluent SS concentration shown in Figure 4 continuously increases from 20 mg/L to 30 mg/L from 11:00 to 23:00, even though the loading to the clarifier is constant. The rising sludge is believed to cause the high secondary effluent SS concentration. It is believed that the settled MLSS accumulates at the bottom of the clarifier because the sludge scraper in the primary rectangular clarifier-type secondary clarifier cannot transfer settled MLSS to the hopper immediately. The opposite direction of settled MLSS and fluid flow at the bottom of the clarifier allows the leftover settled MLSS to accumulate until the loading is subsided. Although the depth of the sludge blanket is not measured, it is believed that the depth increases during that period. As more settled MLSS accumulates, the rate of sludge rising increases, leading to an increase in the secondary effluent SS concentration, as shown in Figure 4.

From 00:00 to 05:00, the loading to the clarifier is reduced, and the hourly average secondary effluent SS concentration drops significantly from 42 mg/L to 24 mg/L. During this period, it is believed that the rate of transfer of settled MLSS to the hopper is higher than the rate of accumulation of settled MLSS at the bottom of the clarifier; therefore, the depth of the settled MLSS blank decreases. It is assumed that the depth is very thin, which reduces the sludge rising. As the loading to the clarifier decreases, the sludge rising is reduced and, eventually, the secondary effluent SS concentration is declined.

**Table 4.** Average hourly secondary effluent SS concentrations and corresponding average hourly inflows, SORs, and SLRs at the secondary clarifier at plant A.

| Sampling Time | Effluent SS Concentration mg/L | Inflow m³/h | SOR m²/m²·h | SLR kg/m²·day |
|---|---|---|---|---|
| 11:00 | 20.5 | 1860 | 21.8 | 115.0 |
| 12:00 | 13.3 | 1858 | 21.8 | 114.9 |
| 13:00 | 18.6 | 1860 | 21.8 | 115.0 |
| 14:00 | 14.4 | 1874 | 22.0 | 115.6 |
| 15:00 | 17.6 | 1861 | 21.8 | 115.0 |
| 16:00 | 24.4 | 1788 | 21.0 | 111.9 |
| 17:00 | 24.4 | 1758 | 20.6 | 110.6 |
| 18:00 | 23.6 | 1744 | 20.4 | 110.0 |
| 19:00 | 23.3 | 1751 | 20.5 | 110.3 |
| 20:00 | 22.3 | 1773 | 20.8 | 111.3 |
| 21:00 | 29.6 | 1806 | 21.2 | 112.7 |
| 22:00 | 27.1 | 1722 | 20.2 | 109.1 |
| 23:00 | 42.4 | 1329 | 15.6 | 92.2 |
| 0:00 | 35.3 | 1079 | 12.6 | 81.5 |
| 1:00 | 36.2 | 992 | 11.6 | 77.8 |
| 2:00 | 36.0 | 920 | 10.8 | 74.7 |
| 3:00 | 30.9 | 824 | 9.7 | 70.6 |
| 4:00 | 26.5 | 740 | 8.7 | 67.0 |
| 5:00 | 24.2 | 649 | 7.6 | 63.1 |
| 6:00 | 30.9 | 667 | 7.8 | 63.8 |
| 7:00 | 36.6 | 1207 | 14.1 | 87.0 |
| 8:00 | 35.9 | 1609 | 18.9 | 104.4 |
| 9:00 | 22.5 | 1793 | 21.0 | 112.3 |
| 10:00 | 16.2 | 1822 | 21.4 | 113.5 |
| Average | 26.3 | 1470.3 | 17.2 | 98.3 |
| Maximum | 42.4 | 1874.4 | 21.0 | 115.6 |
| Minimum | 13.2 | 648.9 | 7.6 | 63.1 |

Note: Average MLSS concentrations from Table 4 is used to calculate the SLR for the respective sampling day.

When the loading to the clarifier is increased in the morning hours, there are two distinct aspects of the secondary effluent SS concentration. Hourly average secondary effluent SS concentration increases to 36 mg/L from 24 mg/L during 05:00–07:00 as SOR and SLR increase from 7.8 to 14.1 $m^3/m^2$·day and from 63.1 to 87.0 $kg/m^2$·day, respectively. However, the hourly average secondary effluent SS concentration decreases from 35.9 to 16.2 mg/L during 08:00–10:00 when the SOR and SLR increase from 18.9 to 21.4 $m^3/m^2$·day and from 104.4 to 113.5 $kg/m^2$·day, respectively. It seems that simply increasing the loading to the clarifier does not bring about a high secondary effluent SS concentration and that the rate of loading increase plays an important role in the secondary effluent SS concentrations. From 05:00 to 07:00, the hourly increases in SOR and SLR are 3.2 $m^3/m^2$·day per hour and 12.0 $kg/m^2$·day per hour, respectively. However, the hourly increases in SOR and SLR from 08:00 to 10:00 are 1.3 $m^3/m^2$·day per hour and 4.6 $kg/m^2$·day per hour, respectively, which are about one-third of the corresponding increase from 05:00 to 07:00. The hourly

average secondary effluent SS concentration increases from 05:00 to 07:00. However, the hourly average secondary effluent SS concentration decreases from 08:00 to 10:00. From these changes, it can be deduced that the high fluid momentum created by the increased loading during 05:00–07:00 pushes the settled MLSS to the end of the clarifier and over the effluent weir. This carried-over MLSS causes a high effluent SS concentration. However, the fluid momentum created by the increased loading during 08:00–10:00 is not sufficiently large to push the settled MLSS to the effluent weir. During this period, it is believed that the incoming MLSS settles well, and sludge rising does not occur because of the small amount of accumulated settled MLSS at the bottom of the clarifier.

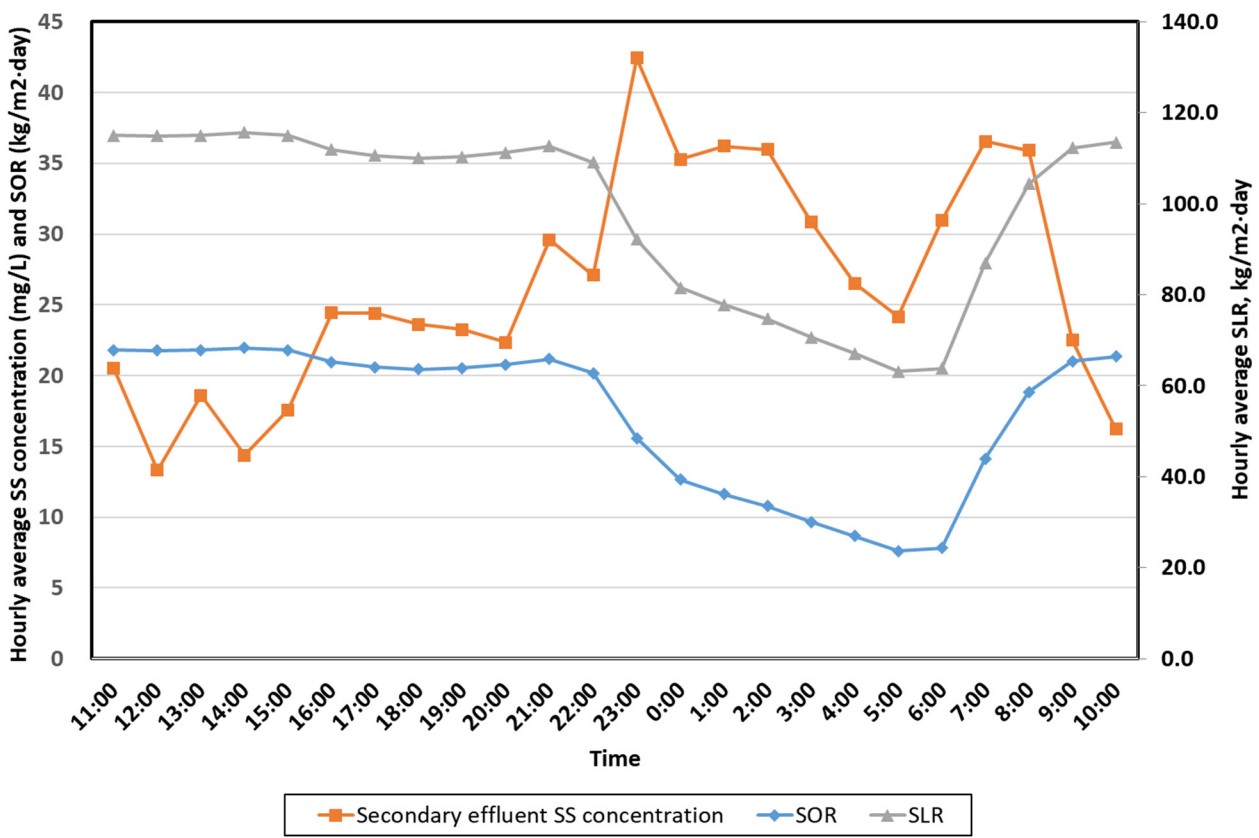

**Figure 4.** Average hourly effluent SS concentrations with average hourly SORs and SLRs at the secondary clarifier at plant A.

For a typical secondary clarifier design, the recommended SOR for biological nutrient removal processes, such as plant A, ranges from 24 to 32 $m^3/m^2 \cdot$day at average flow and from 42 to 64 $m^3/m^2 \cdot$day at peak flow [15]. The recommended design SLR values range from 120 to 192 $kg/m^2 \cdot$day at average daily flow, and 240 $kg/m^2 \cdot$day at peak flow. Because the secondary clarifier at plant A has an SOR of 21 $m^3/m^2 \cdot$day and an SLR of 115.6 $kg/m^2 \cdot$day at peak flow during the study period, this clarifier is assumed to be operated at well under the recommended loading conditions. Also, the average hourly SOR values range from 7.6 to 21.0 $m^3/m^2 \cdot$day with an average value of 17.2 $m^3/m^2 \cdot$day and the average hourly SLR values range from 63.1 to 115.6 $kg/m^2 \cdot$day, with an average value of 98.3 $kg/m^2 \cdot$day. From these values, the ratios of maximum to minimum average hourly SORs and SLRs are 2.9 and 1.8, respectively. The ratio of 2.9 and 1.8 indicate approximately a threefold variation in inflow to the plant, and a twofold variation in loading to the clarifier within a day, respectively. Thus, the diurnal loading of the secondary clarifier at plant A is highly variable. The ratio of maximum to minimum average hourly secondary effluent SS concentrations is 3.2, implying that the variation in SS concentration is higher than that

of both SOR and SLR. From these data, it can be inferred that other factors, such as rising sludge, affect the effluent SS concentration along with the loading variation.

Table 3 shows daily MLSS concentrations in the bioreactor during the sampling period at plant A. The average, maximum, and minimum MLSS concentrations are 3666; 3808; and 3508 mg/L, respectively. Table 5 presents SVI values for the corresponding sampling day. The average, maximum, and minimum SVI values are 210, 213, and 201 mL/mg, respectively. SVI is known to represent the settling characteristics of MLSS [19]. As shown in Table 5, the SVI values are almost constant throughout the sampling period. Since SVI values above 200 mL/g are known to indicate sludge bulking by filamentous microorganisms [20], it can be assumed that MLSS contains filamentous microorganisms. The high SVI values at plant A can be responsible for the high effluent SS concentrations, because the operational SORs and SLRs are much lower than the recommended values.

**Table 5.** SVI values for respective sampling days at plant A.

| Date | SVI mL/mg |
|---|---|
| 9 January 2000 | 210 |
| 10 January 2000 | 208 |
| 13 January 2000 | 210 |
| 14 January 2000 | 209 |
| 21 January 2000 | 210 |
| 22 January 2000 | 213 |
| 4 February 2000 | 201 |
| 5 February 2000 | 212 |
| 13 February 2000 | 209 |
| 14 February 2000 | 211 |
| 18 February 2000 | 213 |
| 19 February 2000 | 211 |
| 27 February 2000 | 210 |
| 28 February 2000 | 209 |
| 5 March 2000 | 209 |
| 6 March 2000 | 207 |
| 12 March 2000 | 209 |
| 13 March 2000 | 210 |
| Average | 210 |
| Maximum | 213 |
| Minimum | 201 |

### 3.2. Experimental and Operational Data from Plant B

Supplementary Table S2 presents the diurnal secondary effluent SS concentrations and corresponding hourly inflow at plant B and Figure 5 shows the diurnal variation in effluent SS concentration during the sampling period at plant B. In contrast to plant A, no significant diurnal secondary effluent SS concentration variation is observed in plant B. Table 6 presents the daily MLSS concentrations and RAS flow ratios during the sampling period at plant B. From this table, SORs and SLRs are calculated based on the total surface area of the clarifier, which is 751.4 m$^2$. Table 7 presents average hourly effluent SS concentrations, inflow, SOR, and SLRs at each sampling hour. Similar to plant A, the daily RAS pumping rate is provided by the plant personnel. Table 8 presents the daily SVI value of MLSS at the aeration tank in plant B during the sampling period.

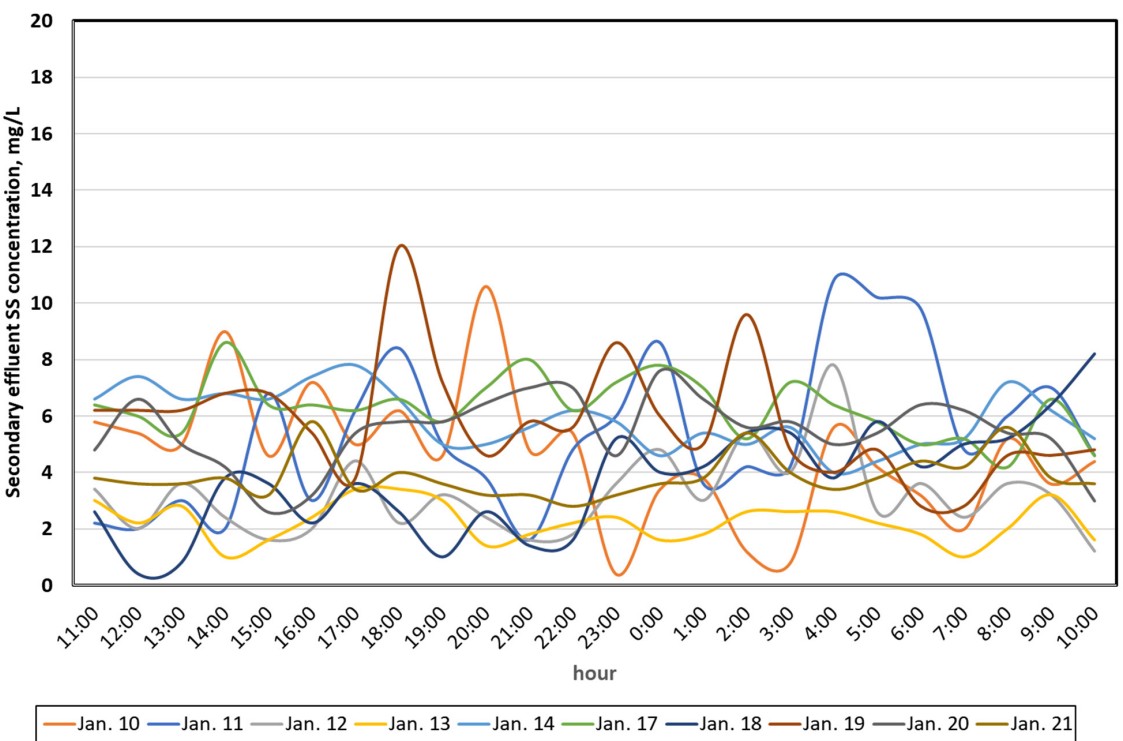

**Figure 5.** Diurnal variation in secondary effluent SS concentrations at the secondary clarifier at plant B during the sampling period.

**Table 6.** Daily inflow, RAS flow, RAS ratio, and MLSS concentrations on each sampling day at plant B.

| Items Date | Inflow m³/Day | RAS Flow m³/Day | RAS Ratio % | MLSS Concentration mg/L | Average MLSS Concentration mg/L |
|---|---|---|---|---|---|
| 10 January 2021 | 11,331 | 10,953 | 96.7 | 3284 | 3290 |
| 11 January 2021 | 10,800 | 11,392 | 105.5 | 3295 | |
| 11 January 2021 | 10,800 | 11,392 | 105.5 | 3295 | 3287 |
| 12 January 2021 | 11,031 | 11,235 | 101.8 | 3278 | |
| 12 January 2021 | 11,031 | 11,235 | 101.8 | 3278 | 3271 |
| 13 January 2021 | 11,630 | 12,009 | 103.3 | 3264 | |
| 13 January 2021 | 11,630 | 12,009 | 103.3 | 3264 | 3258 |
| 14 January 2021 | 11,352 | 11,533 | 101.6 | 3251 | |
| 14 January 2021 | 11,352 | 11,533 | 101.6 | 3251 | 3244 |
| 15 January 2021 | 11,185 | 11,612 | 103.8 | 3237 | |
| 17 January 2021 | 11,300 | 11,786 | 104.3 | 3223 | 3216 |
| 18 January 2021 | 11,004 | 11,301 | 102.7 | 3209 | |
| 18 January 2021 | 11,004 | 11,301 | 102.7 | 3209 | 3203 |
| 19 January 2021 | 10,797 | 11,774 | 109.0 | 3196 | |
| 19 January 2021 | 10,797 | 11,774 | 109.0 | 3196 | 3178 |
| 20 January 2021 | 10,988 | 11,479 | 104.5 | 3160 | |
| 20 January 2021 | 10,988 | 11,479 | 104.5 | 3160 | 3168 |
| 21 January 2021 | 11,055 | 11,820 | 106.9 | 3177 | |
| 21 January 2021 | 11,055 | 11,820 | 106.9 | 3177 | 3172 |
| 22 January 2021 | 11,084 | 11,589 | 104.6 | 3166 | |
| Average | 11,111 | 11,551 | 104.0 | 3229 | 3229 |
| Maximum | 11,630 | 12,009 | 109.0 | 3295 | 3290 |
| Minimum | 10,797 | 10,953 | 96.7 | 3160 | 3168 |

**Table 7.** Average hourly secondary effluent SS concentrations and corresponding average hourly inflows, SORs, and SLRs at the secondary clarifier at plant B.

| Sampling Time Hour | Effluent SS Concentration mg/L | Average Hourly Inflow m³/h | SOR m³/m²·h | SLR kg/m²·day |
|---|---|---|---|---|
| 11:00 | 4.5 | 550 | 17.6 | 115.5 |
| 12:00 | 4.2 | 551 | 17.6 | 115.7 |
| 13:00 | 4.2 | 554 | 17.7 | 116.3 |
| 14:00 | 4.8 | 562 | 17.9 | 118.0 |
| 15:00 | 4.4 | 555 | 17.7 | 116.5 |
| 16:00 | 4.5 | 549 | 17.6 | 115.3 |
| 17:00 | 4.9 | 538 | 17.2 | 112.9 |
| 18:00 | 5.8 | 549 | 17.5 | 115.2 |
| 19:00 | 4.4 | 554 | 17.7 | 116.3 |
| 20:00 | 4.7 | 557 | 17.8 | 117.0 |
| 21:00 | 4.1 | 565 | 18.1 | 118.7 |
| 22:00 | 4.4 | 564 | 18.0 | 118.4 |
| 23:00 | 4.7 | 556 | 17.8 | 116.8 |
| 0:00 | 5.2 | 538 | 17.3 | 113.4 |
| 1:00 | 4.4 | 514 | 16.7 | 108.3 |
| 2:00 | 5.0 | 430 | 13.1 | 90.6 |
| 3:00 | 4.4 | 273 | 8.4 | 57.5 |
| 4:00 | 5.3 | 190 | 6.1 | 40.2 |
| 5:00 | 4.9 | 188 | 6.0 | 39.5 |
| 6:00 | 4.6 | 254 | 7.9 | 53.6 |
| 7:00 | 3.9 | 295 | 9.5 | 62.2 |
| 8:00 | 4.9 | 442 | 14.3 | 93.1 |
| 9:00 | 5.0 | 519 | 16.7 | 109.4 |
| 10:00 | 4.1 | 539 | 17.2 | 113.7 |
| Average | 4.6 | 474 | 15.1 | 99.8 |
| Maximum | 5.8 | 565 | 18.1 | 118.7 |
| Minimum | 3.9 | 188 | 6.0 | 39.5 |

Note: The average MLSS concentrations from Table 8 are used to calculate the SLR for the respective sampling day.

Figure 6 shows the time-series of average hourly secondary effluent SS concentrations, SORs, and SLRs. As displayed in the figure, secondary effluent SS concentrations are relatively constant regardless of SOR and SLR. SOR ranges from 6.0 to 18.1 m³/m²·day, with an average value of 17.2 m³/m²·day. SLR ranges from 39.5 to 118.7 kg/m²·day, with an average value of 63.1 kg/m²·day. From these values, the ratios of maximum to minimum average hourly SOR and SLR values are 3.0 and 3.0, respectively, indicating a threefold variation in SOR and SLR within a day. Although plant B has a similar SOR ratio to plant A, the ratio of SLR is 40% higher in plant B. This implies that the secondary clarifier at plant B receives 40% more variable solid loading than plant A. The overall average hourly secondary effluent SS concentration is 4.6 mg/L and the maximum and minimum average hourly SS concentrations are 5.8 and 3.9 mg/L, respectively. The ratio of maximum to minimum SS concentration is 1.5, which is half the ratio of maximum to minimum SOR and SLR. From this comparison, it can be inferred that the loading variation

to the secondary clarifier at plant B does not have a significant effect on the variation in effluent SS concentration like in plant A.

**Table 8.** SVI values for respective sampling day at plant B.

| Date | SVI mL/mg |
|---|---|
| 10 January 2021 | 272 |
| 11 January 2021 | 273 |
| 12 January 2021 | 274 |
| 13 January 2021 | 275 |
| 14 January 2021 | 275 |
| 15 January 2021 | 276 |
| 17 January 2021 | 276 |
| 18 January 2021 | 277 |
| 19 January 2021 | 278 |
| 20 January 2021 | 279 |
| 21 January 2021 | 279 |
| 22 January 2021 | 280 |
| Average | 276 |
| Maximum | 280 |
| Minimum | 272 |

Table 6 presents the daily bioreactor MLSS concentrations during the sampling period at plant B. As shown in the table, the average, maximum, and minimum MLSS concentrations are 3229; 3295; and 3160 mg/L, respectively. Table 8 presents SVI values for the corresponding sampling days. The average, maximum, and minimum SVI values are 276, 280, and 272 mL/mg, respectively. The SVI values in plant B are relatively constant, similar to SVI values at plant A. However, at plant B, the average MLSS concentration is 11.5% lower and the SVI value is 31.3% higher in comparison to plant A. Although the MLSS concentration in plant B is less than that in plant A, secondary clarifiers at both the plants have similar SLRs as shown in Tables 4 and 7. This anomaly can be explained by the higher RAS ratios at plant B.

The primary rectangular clarifier-type secondary clarifier at Plant A cannot operate with a high RAS ratio. The transportation of settled MLSS to the hopper needs more time than at Gould Type I secondary clarifier, since settled MLSS is moved toward the hopper against the direction of flow on the bottom of clarifier. When the RAS ratio is high at this type of clarifier, the hopper cannot have settled MLSS, and the MLSS with the same concentration of inflow to the clarifier will be RAS. Since RAS is diluted with influent to the bioreactor, the high concentration of MLSS in RAS has to be maintained to keep a desired MLSS concentration in the bioreactor. It is believed that Plant A can only afford the RAS ratios in Table 4 due to the shape of the primary rectangular clarifier-type secondary clarifier. If RAS ratios are higher than those in Table 4, there is no means to sustain the MLSS concentrations presented in Table 3.

Table 9 presents SVI values of Plant A and B. As shown on this table, the MLSS at Plant B has higher SVI values that those in Plant A.

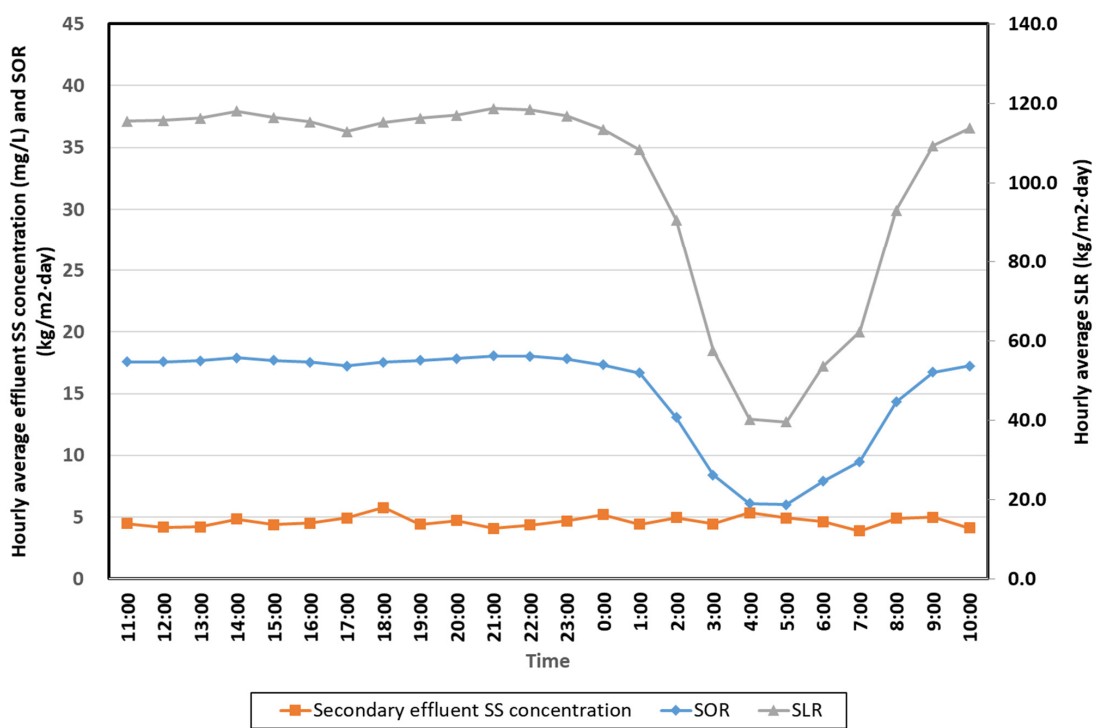

**Figure 6.** Average hourly effluent SS concentration along with average hourly SOR and SLR at the secondary clarifier at plant B.

**Table 9.** SVI values for respective sampling day at plant A and B.

| Plant A | | Plant B | |
|---|---|---|---|
| Date | SVI mL/mg | Date | SVI mL/mg |
| 9 January 2000 | 210 | 10 January 2021 | 272 |
| 10 January 2000 | 208 | 11 January 2021 | 273 |
| 13 January 2000 | 210 | 12 January 2021 | 274 |
| 14 January 2000 | 209 | 13 January 2021 | 275 |
| 21 January 2000 | 210 | 14 January 2021 | 275 |
| 22 January 2000 | 213 | 15 January 2021 | 276 |
| 4 February 2000 | 201 | 17 January 2021 | 276 |
| 5 February 2000 | 212 | 18 January 2021 | 277 |
| 13 February 2000 | 209 | 19 January 2021 | 278 |
| 14 February 2000 | 211 | 20 January 2021 | 279 |
| 18 February 2000 | 213 | 21 January 2021 | 279 |
| 19 February 2000 | 211 | 22 January 2021 | 280 |
| 27 February 2000 | 210 | | |
| 28 February 2000 | 209 | | |
| 5 March 2000 | 209 | | |
| 6 March 2000 | 207 | | |
| 12 March 2000 | 209 | | |
| 13 March 2000 | 210 | | |
| Average | 210 | | 276 |
| Maximum | 213 | | 280 |
| Minimum | 201 | | 272 |

### 3.3. Secondary Effluent SS Concentrations Comparison by Experimental Data of Plants A and B

Tables 4 and 7 show the average hourly effluent SS concentrations, inflows, SLRs, and SORs for the rectangular primary clarifier-type and Gould Type I secondary clarifiers, respectively. From these tables, the overall average hourly SOR of the rectangular primary clarifier-type secondary clarifier is 17.2 m$^3$/m$^2$·day, which 1 s 10.8% higher than that of Gould Type I clarifier, which is 15.1 m$^3$/m$^2$·day. However, the overall average hourly SLR of rectangular primary clarifier type secondary clarifier (98.3 kg/m$^2$·day) is 0.5% lower than that of Gould Type I clarifier (99.8 kg/m$^2$·day). This discrepancy between the SOR and SLR lies in the RAS ratio difference between plants A and B. Because plant B has higher RAS ratios than those of plant A, plant B has a slightly higher overall average hourly SLR than that of plant A.

Since the purpose of a secondary clarifier is the settling of MLSS, the SLR is considered a more important design and operational factor than the SOR [21]. As discussed in Section 3.2, plant B has a higher ratio of maximum to minimum hourly average SLR values than that of plant A, and higher SVI values. The higher ratio can be attributed to a more variable MLSS load. The higher MLSS loading variation and SVI values imply that plant B has poorer operational conditions than plant A. However, plant B always has lower secondary effluent SS concentrations. From the diurnal SLR variation and SVI values, it can be concluded that plant A has better operational conditions for settling MLSS at the secondary clarifier. However, the effluent SS concentrations from plant B are consistently lower than those from plant A, regardless of the higher SLR variation and SVI values. The constantly low effluent SS concentrations from plant B can only be explained by the shape of the secondary clarifier. The concurrent movement of settled MLSS by the sludge scraper to the hopper at the bottom of the Gould Type I clarifier is believed to play an important role in maintaining consistently low effluent SS concentrations.

### 3.4. Computational Fluid Dynamic Simulation Results

CFD simulation is used to observe the movement of fluid and particles in the clarifiers. A simulation package developed by McCorquodale et al. [13] is used to determine the operational differences in the clarifier type. Because this simulation package can handle only two dimensions, an inlet wall with one slot is incorporated in the Gould Type I clarifier simulation for the EDI. For the primary rectangular-type clarifier, a vertical wall with multiple holes, as explained in Section 2.2, is incorporated for simulation. The physical specifications for both clarifiers, which are presented in Tables 1 and 2, are used as input data for each simulation. The empirical settling equation (Equation (1)), which is presented by Takacs et al. [22], is incorporated to express the MLSS settling in the secondary clarifiers. Coefficients of (Equation (1)) are adopted from Lee [23], and are presented in Table 10. To observe the difference in secondary effluent SS concentrations, the same constants are used for both types of clarifier simulations.

$$V_s = V_o \left( e^{-K1 \ (X - X_{min})} - e^{-K2 \ (X - X_{min})} \right) \tag{1}$$

where $V$o = Stokes velocity (settling velocity of a single particle in clear water) (m/h), $K1$ = empirical coefficient for rapidly settling flocs resulting from the fit of batch settling data (m$^3$/kg), $Xmin$ = the concentration of non-settling flocs (kg/m$^3$), and $K2$ = a settling exponent for the poorly settling particles (m$^3$/kg).

Figure 7 shows the simulation results depicting the settled MLSS blanket according to the clarifier type. The settled MLSS blanket depth in the Gould Type I clarifier was much shallower than that in the primary rectangular clarifier-type. Endogenous denitrification within the blanket was susceptible in the MLSS blanket in primary rectangular clarifier-type. As explained previously, sludge rise can occur by endogenous denitrification within the settled MLSS. Because the CFD simulation in this study could not handle biological reactions, sludge rising caused by endogenous denitrification cannot be illustrated. From the operation data and simulation results, it is believed that a high depth of settled MLSS

blanket encourages endogenous denitrification at the bottom of the clarifier, resulting in high effluent SS concentrations in the primary rectangular clarifier-type secondary clarifier.

**Table 10.** Solid settling parameters used in computerized fluid dynamics (CFD) simulation.

| Elements | Values | | Remarks |
|---|---|---|---|
| | **Plant A** | **Plant B** | |
| MLSS (kg/m$^3$) | 3.400 | 3.300 | |
| ESS (kg/m$^3$) | 0.005 | | *Xmin* in Equation (1) |
| $V$o (m/h) | 14.718 | | |
| $K$1 (m$^3$/kg) | 0.484 | | |
| $K$2 (m$^3$/kg) | 9.500 | | |

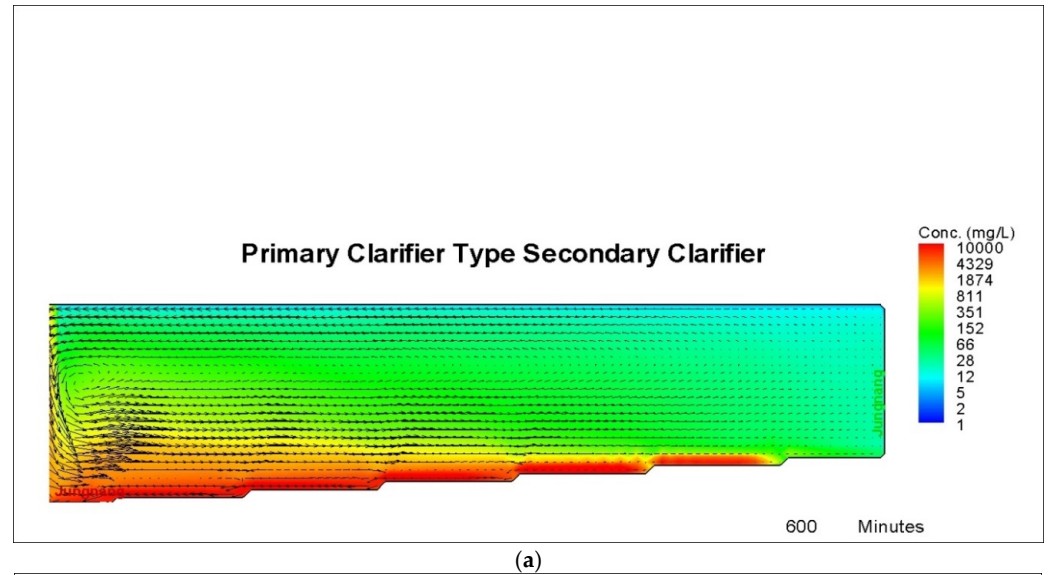

(**a**)

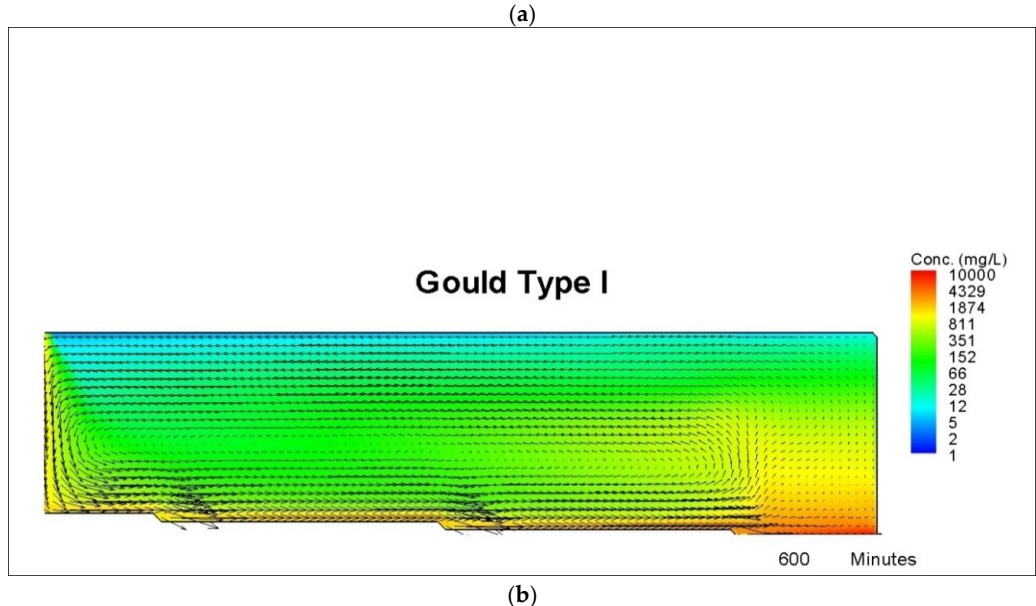

(**b**)

**Figure 7.** (**a**) Two-dimensional SS concentration contour profile for the secondary clarifier at plant A. (**b**) Two-dimensional SS concentration contour profile for the secondary clarifier at Plant B.

Figure 8 shows the simulation results of hourly effluent SS concentrations profile at primary rectangular-type secondary clarifier along with hourly SORs and SLRs. The

secondary effluent SS concentrations are very sensitive to the inflow loading as shown in Figure 4, which presents the experimental data. In the simulation, SS concentrations vary from 11 to 45 mg/L; however, there is no sudden rise in SS concentration under constant loading conditions as observed in experimental data (Figure 3). It is believed that SS concentration peaks in the experimental data are caused by sludge rising which CFD cannot simulate. However, the simulation results show that the falling and rising SS concentrations are proportional to the loading in the secondary clarifier. It should be noted that the effluent SS concentrations in the Gould Type I clarifier are irrelevant to influent loading, as shown in Figure 9, and this is in good agreement with the experimental data, as shown in Figure 6. Comparing the simulation and experimental data, it is observed that the simulation results represent the experimental data at the Gould Type I clarifier well.

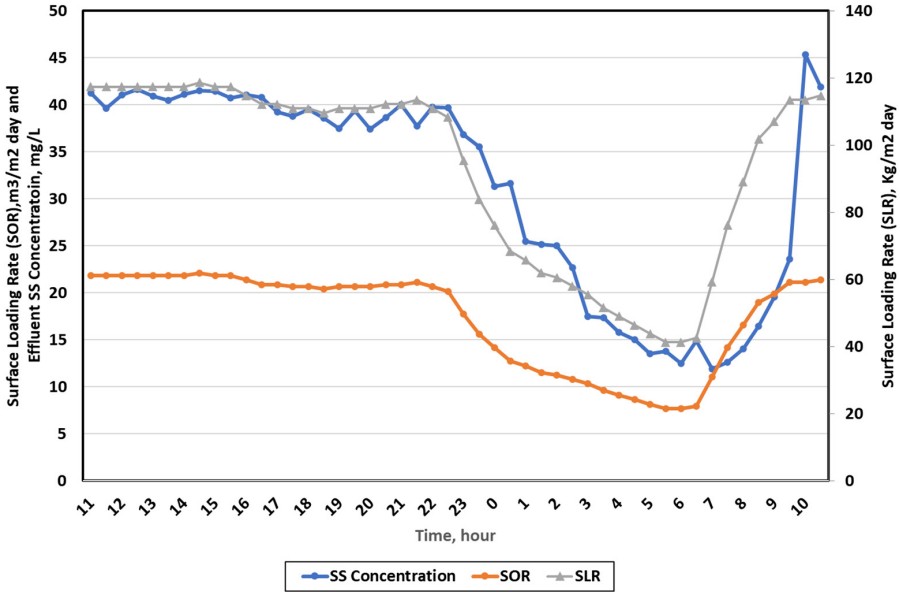

**Figure 8.** Simulation results of hourly secondary effluent SS concentrations and corresponding hourly SORs and SLRs at the secondary clarifier at plant A.

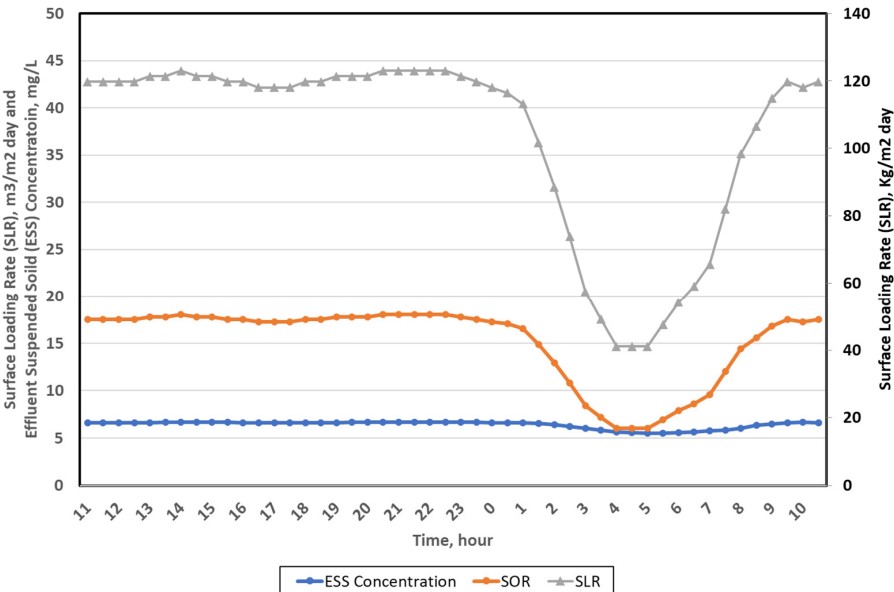

**Figure 9.** Simulation results of hourly secondary effluent SS concentrations and corresponding hourly SORs and SLRs at the secondary clarifier at plant B.

Figure 10 presents simulation results as normalized effluent SS concentrations and normalized SLRs for the respective clarifier type. For normalized SLRs, Plant A and B have ranges from 0.44 to 1.28 and from 0.40 and 1.19, respectively. For normalized SS concentrations, Plant A and B have ranges from 0.38 to 1.44 and from 0.85 to 1.04, respectively. Simulation results show that the difference in normalized SS concentrations at Plant B is small compared with those at Plant B, although normalized SLRs are similar for both types. Simulation shows that effluent SS concentration at Gould Type I clarifier is very resilient against SLR variation.

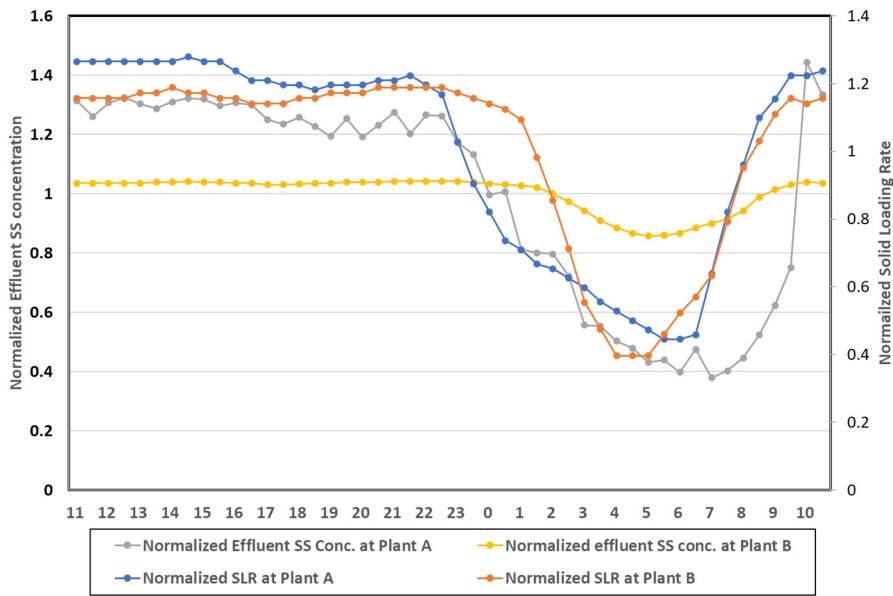

**Figure 10.** Normalized hourly effluent SS concentrations and SLRs at Plants A and B.

## 4. Conclusions

To determine the difference in performance between secondary clarifier types, primary rectangular clarifier-type and Gould Type I secondary clarifiers were studied. Effluents from both types of secondary clarifiers were collected hourly for two weeks to measure SS concentrations and CFD simulations for both types were also conducted. In addition, clarifier operation data were evaluated to determine the effect of loading variation in secondary clarifiers on effluent SS concentrations. Based on this evaluation, the findings are summarized as follows:

SS concentrations in effluents from Gould Type I were relatively constant, although the variation in solid loading in this clarifier was greater than that in primary rectangular clarifier-type secondary clarifiers. For Gould Type I clarifier, the cocurrent flow of fluid and movement of settled MLSS on the bottom of clarifier was believed to resist the high variation in the solid load rate ratio (maximum to minimum SLR ratio of 3.0). The ratio of maximum to minimum SLR was 1.8 for the primary rectangular clarifier-type (40% lower than that of Gould Type I). The maximum ESS concentration in Gould Type I was 12.0 mg/L, whereas the maximum ESS concentration in primary rectangular-type clarifier with a similar average SLR was 130.0 mg/L. These experimental data prove that the Gould Type I clarifier produces effluents with a constantly low SS concentration under highly variable loading conditions.

As shown in the experimental data, the CFD simulation shows highly fluctuating and high effluent SS concentrations in the primary rectangular clarifier-type secondary clarifier. However, the Gould Type I clarifier shows consistently low effluent SS concentrations. The simulation showed a thick settled MLSS blanket at the bottom of the clarifier in the primary rectangular clarifier-type secondary clarifier. This thick blanket is believed to

have undergone endogenous denitrification, which caused sludge to rise. Sludge rising is believed to cause extremely high ESS concentrations, as shown in the experimental data.

Primary rectangular clarifier-type secondary clarifiers are commonly adopted in South Korea. Many treatment facilities with this type of secondary clarifier experience fluctuating and high secondary effluent SS concentrations during winters. From the observations and simulation results, it is highly recommended to convert the existing primary rectangular clarifier-type clarifiers to Gould Type I to produce the effluent having consistently low SS concentrations.

**Supplementary Materials:** The following are available online at https://www.mdpi.com/article/10.3390/w14101577/s1, Table S1: Diurnal secondary effluent SS concentrations and inflows during sampling period at plant A, Table S2: Diurnal secondary effluent SS concentrations and inflows during sampling period at Plant B during.

**Funding:** This research was funded by Kyonggi University Research Grant grant number 2019-001.

**Institutional Review Board Statement:** Not applicable.

**Informed Consent Statement:** Not applicable.

**Acknowledgments:** This work was supported by a Kyonggi University Research Grant (2019). The author would like to thank Kyoon Bum Choi, Seung Won Seo, and Sung-Chul Jung at the Department of Environmental Energy Engineering of Kyonggi University and the wastewater treatment plant staff who provided the operation data. Also, I would like to acknowledge the late Randal Wayne Samstag who inspired me to pursue this study.

**Conflicts of Interest:** The authors declare no conflict of interest.

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
