# Peer review of "Comparison of Effluent Suspended Solid Concentrations from Two Types of Rectangular Secondary Clarifiers"

_water, doi:10.3390/w14101577_

Round 1

Reviewer 1 Report

Must be complete the Back Matter:

Funding

Data Availability Statement

Conflicts of Interest

The References chapter does not comply with the journal requirements.

Author Response

Point 1: The References chapter does not comply with the journal requirements.

Response 1: The author reviews reference and revised the format according to the journal requirement.

Reviewer 2 Report

Overall, this is a very well written article with sound principles and data analysis. I recommend that this article be accepted with very few clarifications as requested below. 

1) General Comment - The reported work uses CFD simulations, to compare performance of 2 existing secondary clarifier technologies in South Korea activated sludge system in terms of clarified effluent Suspended Solids (SS) concentration.

Simulated results suggest to replace existing secondary clarifier type from primary rectangular clarifier type (used in South Korea plant) into Gould type 1 clarifier, since the latter system can handle varying load while producing lower SS effluent consistently.

2) General Comment - 

Although the plant capacity for Plant A (47,000 m3/day max) is almost 3.5 times larger than Plant B (14,000 m3/day max), their corresponding unit secondary clarifier dimensions are almost identical including the clarifier depth, hence the operational data collected for evaluation and simulation are valid and acceptable. 

Data are clearly presented for individual Plant A and B performance. However, they should consider to adding 1 more chart/ diagram to show the normalized combined simulated data to see the difference between 2 plants.

3) Conclusions are well supported by the data because although the authors are not able to include biological reaction (endogenous denitrification) of MLSS blanket, the onsite collected data of SVI and MLSS is sufficient to justify the effluent SS concentration and for performance comparison.

4) Will Gould Type 1 with porous inlet plate work better (better inflow distribution) and with lesser maintenance hassle than EDI? Please clarify.

Author Response

Point 1: 2) General Comment – 

Data are clearly presented for individual Plant A and B performance. However, they should consider to adding 1 more chart/ diagram to show the normalized combined simulated data to see the difference between 2 plants.

Response 1: The normalized combined simulated data are presented in Figure 10 in page 27 in revised manuscript. The explanation for this figure is described between lines numbers 472 to 478, page 23 in revised manuscript.

Figure 10. Normalized hourly effluent SS concentrations and SLRs at Plants A and B. 

Figure 10 presents simulations results as normalized effluent SS concentrations and normalized SLRs for respective clarifier type. For normalized SLRs, Plant A and B have the ranges from 0.44 to 1.28 and from 0.40 and 1.19, respectively. For normalized SS concentrations, Plant A and B have the ranges from 0.38 to 1.44 and from 0.85 to 1.04, respectively. Simulation results show that the difference of normalized SS concentrations at Plant B is small compared with those at Plant B, although normalized SLRs are similar for both types. Simulation shows that effluent SS concentration at Gould Type I clarifier is very resilient against SLR variation.

Point 2: 4) Will Gould Type 1 with porous inlet plate work better (better inflow distribution) and with lesser maintenance hassle than EDI? Please clarify.

Response 2: The author explains how EDI is working and the reason why EDI has to be used in Gould Type I is described between line numbers 168 to 180, page 6 in revised manuscript.

When MLSS flow enters EDI, the plate at the end of EDI blocks the flow and MLSS spreads into four (4) outlet wings which are attached at the end plate of EDI. EDI converts MLSS flow direction from horizontal to vertical. Through this conversion, MLSS is spread along with the front wall of clarifier. Normally there are three (3) EDIs at each clarifier. The outlet wings of each EDI are aligned and MLSS from outlets at each EDI is collided. Two (2) EDIs located near side walls of clarifier send MLSS toward side walls through two (2) outlet wings of each EDI. EDI converts flow direction and spreads MLSS throughout whole cross section of clarifier. Due to high velocity within EDI, no operational problem such as MLSS deposit is reported. 

While EDI spreads MLSS along with the front wall, porous inlet plate injects MLSS into clarifier through holes. If porous inlet plate is used instead of EDI, high horizontal velocity is expected at Gould Type I clarifier. For Gould Type I clarifier, MLSS must be entered through EDIs.

Reviewer 3 Report

Lee et al studied the Comparison of effluent suspended solid concentrations from 2 two types of rectangular secondary Clarifiers. The study is comprehensive, complete, and interesting. I suggest the following additional suggestions to improve the quality of the paper. 

Comments:

  1. Please consider changing primary rectangular clarifier-type to “basic rectangular”
  2. Page 1 Line 10: Please make it clear the function of clarifier.
  3. Page 3 Line 102- 104: Please explain why the author took sample different time between Plant A and Plant B. If the author tries to compare those Plant, sample should take at the same time, so there is no anomaly gap in the conclusion.
  4. Please re-organize figure 1 and 2. Place the affluent part in the same position (all in the right or left part)
  5. Its better to put all table of experiment data in supplementary data. Only table of analyzed data in the manuscript, especially table comparison on Plant A and B based on all parameter, not only based on design of clarifier.
  6. In page 9, please check again the value of max, min, and average between table and text. There is some different value (Page 9 Line 266 and 268)
  7. Page 16 Lin3 337: How RAS can be anomaly please explain more detailed.
  8. Page 24 Line 374: Please make in one table comparison.
  9. Focused on effluent point in the figure 7. It shows in that point SS concentration in Plant A is lower than Plant B (right-up corner). Also, the effluent part area in Plant B there more higher SS concentration in the mid to up (Right region). This simulation is not showing same as with the conclusion of the experiment data.

Author Response

Point 1: Please consider changing primary rectangular clarifier-type to “basic rectangular”.

Response 1: The author thinks “Basic Rectangular” is too broad. The author would like to emphasize the difference of secondary rectangular clarifier type. The type of secondary rectangular clarifier using in S. Korea is identical as primary rectangular clarifier-type except for one facility which the author refers. The conclusion of this paper is “not to use primary rectangular clarifier-type for secondary clarifier”.  The author would like specify the type of clarifier in the paper. I hope the reviewer understands author’s intention.

Point 2: Page 1 Line 10: Please make it clear the function of clarifier.

Response 2: The author added the function of clarifier.

“rectangular clarifiers to settle mixed liquor suspended solids (MLSS)”

Point 3: Page 3 Line 102- 104: Please explain why the author took sample different time between Plant A and Plant B. If the author tries to compare those Plant, sample should take at the same time, so there is no anomaly gap in the conclusion.

Response 3: Two plants are not located at the same site and each plant is operated independently. In this paper, the author would like to see how the type of rectangular clarifier gives an effect on clarifier effluent suspended solid concentrations. The author collected operational data and compared hourly effluent suspended solid concentrations for both types of clarifiers. The author did not assume same operation conditions and used real operation data to see the performance difference according to clarifier type based on loading to each clarifier type. This is why samples from both plants were not taken at the same.

Point 4: Please re-organize figure 1 and 2. Place the affluent part in the same position (all in the right or left part)

Response 4: The author changed the Figure 2

Point 5: Its better to put all table of experiment data in supplementary data. Only table of analyzed data in the manuscript, especially table comparison on Plant A and B based on all parameter, not only based on design of clarifier.

Response 5:  The author changed. I agree with reviewer’s comments. The author allocates the tables with experimental data as supplementary Table 1 and 2 at the end of revised manuscript.

Point 6: In page 9, please check again the value of max, min, and average between table and text. There is some different value (Page 9 Line 266 and 268).

Response 6:  The values in Table are correct. The author changed values according to tables.

Point 7: Page 16 Lin3 337: How RAS can be anomaly please explain more detailed.

Response 7:  The author explained in detail between line numbers 355 to 365, page 15 in revised manuscript.

Primary rectangular clarifier-type secondary clarifier at Plant A cannot operated with high RAS ratio. The transportation of settled MLSS to the hopper needs more time than at Gould Type I secondary clarifier, since settled MLSS is moved toward the hopper against the direction of flow on the bottom of clarifier. When RAS ratio is high at this type of clarifier, the hopper cannot have settled MLSS and the MLSS with same concentration of inflow to the clarifier will be RAS. Since RAS is diluted with influent to bioreactor, the high concentration of MLSS in RAS has to be maintained to keep desired MLSS concentration in bioreactor. It is believed that Plant A can only afford the RAS ratios in Table 4 due to the shape of Primary rectangular clarifier-type secondary clarifier. If RAS ratios are higher than those in Table 4, there is no means to sustain MLSS concentrations presented in Table 3.

Point 8: Page 24 Line 374: Please make in one table comparison

Response 8: The author made the Table 9 , page 21 in revised manuscript for comparison.

Table 9 presents SVI values of Plant A and B. As shown on this table, the MLSS at Plant B has higher SVI values that those in Plant A. 

Point 9: Focused on effluent point in the figure 7. It shows in that point SS concentration in Plant A is lower than Plant B (right-up corner). Also, the effluent part area in Plant B there more higher SS concentration in the mid to up (Right region). This simulation is not showing same as with the conclusion of the experiment data.

Response 9:  In this paper, dynamic CFD simulation were carried out and the results were presented as Figure 8 and 9. The Figure 7 is the snapshots of these dynamic simulations. The point of effluent at primary rectangular clarifier-type is the end of clarifier as shown on Figure 1, while Gould Type 1 clarifier has the finger weir which effluent weir is horizontally intruded into the clarifier as shown on Figure 2. This means that effluent is collected a part of longitudinal length of Gould Type I clarifier. When this is considered, it is very difficult to claim that the effluent at Gould Type I has higher SS concentration. Also, the color which represents the SS concentration has a range. It is very difficult to distinguish color which represents 40 mg/L of effluent SS concentration at primary rectangular clarifier-type and 6 mg/L of of effluent SS concentration at Gould Type I. Effluent SS concentration data are generated when the simulation is finished. These data are used for Figure 8 and 9. For this paper, the author performed dynamic simulation for both types of clarifiers and presented the data in the manuscript. The author did his best for simulations and experiments, and concluded that simulation was well agreed with experimental data.

Round 2

Reviewer 3 Report

The authors have revised the manuscript sufficiently enough to be accepted for publication.  Therefore, i recommend the paper to be accepted and can be published in the present form.